# HR-GLDD: A globally distributed dataset using generalized DL for rapid landslide mapping on HR satellite imagery

Sansar Raj Meena [1], Lorenzo Nava [1], Kushanav Bhuyan [1], Silvia Puliero [1], Lucas Pedrosa Soares [2], Helen Cristina Dias [2], Mario Floris [1], Filippo Catani [1*]

1. Machine Intelligence and Slope Stability Laboratory, Department of Geosciences, University of Padova, 35129 Padua, Italy
2. Institute of Energy and Environment, University of São Paulo, São Paulo 05508-010, Brazil

* Correspondence: sansarraj.meena@unipd.it

**Abstract:**

Multiple landslide events occur often across the world which have the potential to cause significant harm to both human life and property. Although a substantial amount of research has been conducted to address mapping of landslides using Earth Observation (EO) data, several gaps and uncertainties remain when developing models to be operational at the global scale. The lack of a high resolution globally distributed and event-diverse dataset for landslide segmentation poses a challenge in developing machine learning models that can accurately and robustly detect landslides in various regions, as the limited representation of landslide and background classes can result in poor generalization performance of the models. To address this issue, we present the high-resolution global landslide detector database (HR-GLDD), a high resolution (HR) satellite dataset (PlanetScope, 3 m pixel resolution) for landslide mapping composed of landslide instances from ten different physiographical regions globally: South and South-East Asia, East Asia, South America, and Central America. The dataset contains five rainfall-triggered and five earthquake-triggered multiple landslide events that occurred in varying geomorphological and topographical regions in the form of standardized image patches containing four PlanetScope image bands (red, green, blue, and NIR) and a binary mask for landslide detection. The HRGLDD can be accessed through this link https://doi.org/10.5281/zenodo.7189381 (Meena et al., 2022a).. HR-GLDD is one of the first dataset for landslide detection generated by high resolution satellite imagery which can be useful for applications in artificial intelligence for landslide segmentation and detection studies. Five state of the art deep learning models were used to test the transferability and robustness of the HR-GLDD. Moreover, three recent landslide events were used for testing the performance and usability of the dataset to comment on the detection of newly occurring significant landslide events. The deep learning models showed similar results for testing the HR-GLDD in individual test sites thereby indicating the robustness of the dataset for such purposes. The HR-GLDD can be accessed open access and it has the potential to calibrate and develop models to produce reliable inventories using high resolution satellite imagery after the occurrence of new significant landslide events. The HR-GLDD will be updated regularly by integrating data from new landslide events.

1. Introduction

With the increasing impacts of climate change, increased urbanization, and anthropogenic pressure in recent years, the risk from hazards to population, infrastructure, and essential life services has exacerbated. Landslides are quite ubiquitous and account for approximately 4.9% of all the natural disasters and 1.3% of the fatalities in the world (EM-DAT, 2018). Induced by natural (earthquakes, volcanic eruptions, meteorological events) and anthropogenic triggers (slope modifications, mining, landscape engineering), the increase in

the stress of slope materials causes landslides, which can harm numerous elements at risk.
Landslides occur heterogeneously in many parts of the world including the Central and South
Americas, the Caribbean islands, Asia, Turkey, European Alps, and East Africa (Froude &
Petley, 2018). In the past 15 years, we have seen a high number of events that have
inadvertently led to the failure of thousands of slopes and causing damage to essential linear
infrastructures and population. Some recent examples are Wenchuan, China (2008),
Kedarnath, India (2013), Kaikoura, New Zealand (2016), Jiuzhaigou, China (2017), Dominica
(2017), Porgera, Papua New Guinea (2018), Hokkaido, Japan (2018), Belluno, Italy (2018),
Haiti (2021), Sumatra, Indonesia (2022).
These examples indicate that landslide occurrences will probably continue to increase in the
short and medium term; therefore, an effective capability of rapid mapping is required to map
future event-based landslides. In recent years, state-of-the-art research has been conducted
to better understand the impact of natural hazards such as landslides and the cascading
effects on the elements-at-risk. A critical understanding of these complex processes begins
with the onset of mapping slope failures. This information about the failed slopes is attributed
as records and is documented in a "landslide inventory". Landslide inventories include
information on the spatial location and extent of the landslides and, if available, also crucial
information about 1) the time of occurrence, 2) the triggering event that led slopes to fail, 3)
the typology of the landslides based on the accepted standard classifications like (Cruden &
Varnes, 1996) and (Hungr et al., 2014), and 4) the volume of the failure. However, regarding
rapid mapping of recently occurred landslides, information about the spatial location,
distribution, and intersection with affected elements-at-risk are important. , and 4) the volume
of the failure. However, regarding rapid mapping of recently occurred landslides, information
about the spatial location, distribution, and intersection with affected elements-at-risk are
important.
When it comes to detecting and mapping landslides over remotely sensed images, it is safe
to say that a lot of the current literature in the past couple of years has devised and spent time
employing artificial intelligence (AI) models to map landslides automatically, arguably, with
good results. These AI models can classify remote sensing images to denote where the
landslides are present in the analysed images. However, the core prerequisite for employing
AI models is a reliable dataset to be used for training. Recent studies have only focused on
mapping landslides with AI but at scales that are small or regional while also claiming that the
proposed models can cater towards rapid mapping of landslides at any given time, location
and scale (Liu et al., 2022; Meena et al., 2022b; Nava, Bhuyan, et al., 2022; Nava, Monserrat,
et al., 2022; Soares et al., 2022a; Tang et al., 2022; Yang et al., 2022; Yang & Xu, 2022).
However, seldom has been the case where truly an approach has been taken to map
landslides outside the regions where the models are initially trained on, and also towards
actually applying the proposed models in capturing and mapping event-based landslides that
has recently occurred. Some other works at collectively detecting and mapping landslides of
different countries have been attempted by (Prakash et al., 2021) and (Ghorbanzadeh et al.,
2022), which showcases the power of employing AI at mapping landslides. Recently, Bhuyan
et al. (2023) made some strides at mapping landslides at larger spatiotemporal scales to
provide multi-temporal inventories of some famous events but more experiments in to explore
other geographical contexts are required. The core of the mentioned studies also heavily relies
on the availability of quantity and quality data for training an AI model. The accessibility of
such data can 1) allow a model to identify landslides that were caused by different types of
triggers (logically leading to the detection of different types of landslides), 2) to map landslides
in different parts of the world that vary geomorphologically, and 3) the applicability of the model
at mapping newly occurring landslides triggered by events in recent times. The contemporary
works of the current literature brings about a critical discussion about the availability and
accessibility of comprehensive and adequate data to effectively train models to detect
landslides. Both (Prakash et al., 2021) and (Ghorbanzadeh et al., 2022) have used open-
source Sentinel-2 imageries for multi-site landslide detection however, considering the fact
that the spatial resolution is 10 metres, a lot of small landslides are missed out or not
accurately captured (Meena et al., 2022b). The latter sampled data from 4 different
areas/events Sentinel-2 imagery (four bands at 10 meters spatial resolution, six at 20, and
three at 60) and combined it with DEM derived data from ALOS-PALSAR. The dataset we
propose, instead, is sampled from 10 different areas/events and uses 3 meters spatial
resolution imagery. Sampling from more areas can provide a more diverse representation of
both landslide and background classes, which can improve the robustness of the model when
applied to different regions. Moreover, a dataset with more diversity is likely to generalize
better to new unseen data than one with limited diversity, making it more suitable for real-
world deployment. Sampling from 10 areas also provides better coverage of the geographical
region, reducing the risk of missing important features or patterns. Higher spatial resolution
imagery captures more detail, allowing for more accurate identification and segmentation of
landslide features. It also allows obtaining a more detailed view, which can be useful to identify
small landslides or details that may be difficult to see in lower resolution imagery. Moreover, it
can provide more context for the location, helping to better understand the environment and
the relationships between different objects and features. Therefore, the increased detail can
result in improved accuracy when classifying features and objects, reducing the risk of
misclassification.
To effectively and rapidly map landslides after an event, it is required first to determine the
spatial extent of the affected areas. Collecting this data is frequently hazardous since it
involves individuals on the ground investigating landslides first hand during or immediately
after the event. With the increased availability of satellite imagery, this task has the potential
to be completed not only remotely but also automatically through the use of powerful deep
learning algorithms. Currently, adequate high-resolution satellite imagery of landslides is not
widely available. To depict the complex and dynamic nature of the landslides, significant
amounts of images must be provided. To this purpose, we present high-resolution global
landslide detector database (HR-GLDD), a large-scale satellite image dataset with assembled
landslide inventories. The database currently houses 10 geographical areas and 3 recently
transpired events (see Figure 1), and we plan to constantly update this database with newer
events.

2.  Study areas
The study areas were chosen based on the variety of triggering events that resulted in the
occurrence of the landslides. Because of the availability of VHR archived Planet Scope
imageries after 2016, the most significant landslide events were considered. The
geomorphological diversity of the study sites results in a collection of complex landslide
phenomenon. We selected the imageries based on the availability of cloud-free conditions in
the areas and examined globally archived satellite remote sensing imageries from Planet
Scope from the years between 2017 and 2022 (Table 1). We selected 8 study sites across the
globe to assemble the database (see figure1). To further test the generalization capabilities of
the models trained on the proposed dataset, we choose three recently occurred events: co-
seismic landslides in Haiti (August, 2021) and rainfall-induced landslides in Indonesia
(February, 2022) and Democratic Republic of Congo (April, 2020) (Meena et al., 2022a).

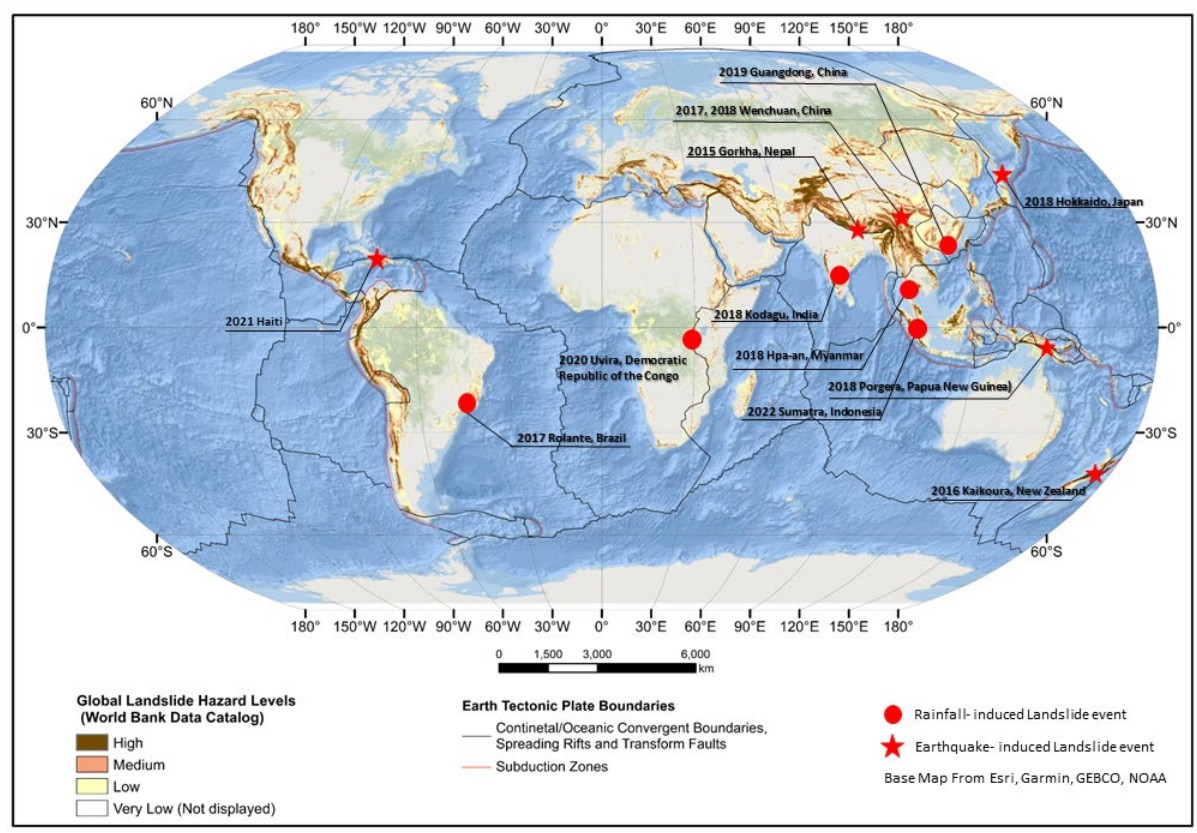


Figure 1: Collection of rainfall- and earthquake-induced landslide events present in the HR-
GLDD.

## 2.1. Porgera, Papua New Guinea
Papua New Guinea (PNG), located on the Australian continent, is the eastern half of the New
Guinea island. This region, characterized by active volcanos, earthquakes, and steep slopes
with elevations up to ~4.400 m.a.s.l., is part of the Pacific Ocean's 'Ring of Fire'. The geological
and tectonic makeup divides the island into four tectonic belts: Stable platform, Fold Belt,
Mobile Belt, and Papuan Fold and Thrust Belt (Tanyaş et al. 2022). Particularly in the east,
where PNG lies, there exists an accreted Paleozoic structure known as the Tasman Orogen
(Hill and Hall, 2003). Due to these unique geotectonic conditions, the area is frequently
affected by landslides associated with the occurrence of earthquakes (Tanyaş et al. 2022). On
February 25, 2018, a severe earthquake struck the southern region of the Papuan Fold and
Thrust belt (central highlands of PNG), reaching a magnitude of Mw 7.5. This event, the
highest magnitude in the region in the past century (Wang et al. 2020), caused significant
damage to buildings and energy structures while also triggering a massive number of
landslides. This 2018 earthquake in PNG instigated over 200 landslides across the affected
area, resulting in numerous fatalities and substantial infrastructural damage. The primary
causes for these landslides were the intense ground shaking and the region's steep
topography. Additional influential factors included soil characteristics, rainfall, and vegetation
cover. A deep understanding of these contributing elements can significantly enhance
landslide hazard assessments and aid in reducing future risk (Dang et al., 2020; Xu et al.,
2020). Characteristics of the landslides included high relief, steep slopes, and weak lithology.
An impressive number of 11,600 landslide scars were recorded post-event, with more than
half surpassing an area of 50,000 m² (Tanyaş et al. 2022). Given these realities, effective
strategies for managing landslide hazards in such high-risk areas must be developed and
implemented.

### 2.2. Kodagu, India

Kodagu district is located in the Karnataka state, Western Ghats, India. The area is
characterized by elevations approximately between 50 and 1.750 m a.s.l., metamorphic rocks
(e.g., amphibolite, gneiss, and schist), steep slopes, high annual precipitation of about 4000
mm, and the presence of croplands (e.g., coffee, rice, and spices) (Jennifer and Saravan,
2020; Meena et al. 2021). In August 2018, a rainfall-induced high magnitude mass movement
event occurred in Kodagu, the primary landslide type triggered was debris flow (Meena et al.
2021). A total of 343 landslides were recorded, including mudflows, rock falls, and debris flows
(Meena et al. 2021). The event resulted in several damages to land resources, properties, and
loss of human lives (Martha et al. 2018; Jennifer and Saravan, 2020).

### 2.3. Rolante, Brazil

The Rolante river catchment study area is located in the Rio Grande do Sul state, southern
Brazil. The region being part of the Serra Geral geomorphological unit, has elevations up to
~1.000 m.a.s.l. (Uehara et al. 2020). Moreover, is characterized by the presence of basaltic
rocks and sandstones, and annual precipitation thresholds between 1700 and 2000 mm
(Uehara et al. 2020, Soares et al. 2022). On 5 January 2017, a high magnitude rainfall-induced
mass movement event was triggered, and 308 landslides were registered (Gameiro et al.
2019; Quevedo et al. 2019), resulting in several damages to the Rolante municipality.

### 2.4. Tiburon Peninsula, Haiti

The Tiburon Peninsula study area is located in the western part of the Hispaniola island (Haiti)
with elevation up to 2300 m. a.s.l. Tiburon Peninsula, mainly consists of volcanic rocks such
as basalts and sedimentary rocks, namely limestones (Harp et al., 2016). The annual
precipitation of the area is more than 1600 mm (Alpert, 1942; USAID, 2014). On 14 August
2021, Tiburon Peninsula was struck by a Mw 7.2 earthquake, which was followed by several
aftershocks. The strongest one (Mw 5.7) occurred on 15 August 2021. Two days after the
mainshock the area was hit by the intense Tropical Cyclone Grace. The combination of the
two events triggered thousands of landslides (Martinez et al., 2021) in the Pic Macaya National
Park located in western part of the peninsula.

### 2.5. Rasuwa, Nepal

The study area is located in the Rasuwa district (central Nepal) in the higher Himalayas with
altitudes ranging from 904 to 3267 m. a.s.l and annual average precipitation of 1800-2000 mm
(Karki et al., 2016),The geology includes Proterozoic metamorphic rocks such as amphibolite,
gneiss, and schist (Tiwari et al., 2017). The area was struck by the Mw 7.8 Gorkha earthquake
on 25 April 2015. The intense seismic sequence produced at least 25,000 landslides (Roback
et al., 2018).

### 2.6. Hokkaido, Japan

The Hokkaido study area is in northern Japan and has a high presence of croplands. The area
is characterized by elevations between 50 and 500 m a.s.l., the geology is composed of
Neogene sedimentary rocks, formed by the accumulation of numerous layers formed by
materials ejected by the Tarumai volcano from several events over the years (Yamagishi and
Yamazaki, 2018; Zhao et al. 2020; Koi et al. 2022). A severe earthquake hit the Hokkaido
Iburi-Tobu area in Japan on September 6th, 2018. The earthquake registered a magnitude of
6.7 according to the Japan Meteorological Agency (JMA) and its epicenter was at 42.72° North

and 142.0° East (Yamagishi and Yamazaki, 2018), located along the southern frontier of Hokkaido. The event triggered thousands of landslides (~7059) in a concentrated area of 466 km² (Zhao et al. 2020) and was responsible for 36 deaths (Yamagishi and Yamazaki, 2018).

### 2.7. Wenchuan, China

The study area is in the Longmenshan region at the eastern margin of the Tibetan Plateau, China. The location is characterized by high elevations up to 7.500 m a.s.l., the geology consists of lithological units from the Mesozoic, Jurassic, Cretaceous, Paleozoic, Precambrian formations and three types of Quaternary sedimentary units (Qi et al. 2010; Gorum et al. 2011). The area is constantly affected by earthquake-induced landslides over the years (e.g., 2017, 2018, 2019, 2021). The 2008 Wenchuan event is one of the most destructive events of mass movements related to earthquakes in the region (Fan et al. 2018). The Wenchuan earthquake hit a magnitude of Mw 7.9. It was responsible for triggering nearly 200.000 landslides (Xu et al. 2014), besides missing, injured, and thousands of human fatalities in a total area of 31,686.12 km² (Qi et al. 2010).

### 2.8. Sumatra, Indonesia

The investigated area is Mount Talamau (2912 m) which is a compound volcano located in West Pasaman Regency, West Sumatra Province, Indonesia. Geologically, the volcano consists of andesite and basalt rocks belonging to Pleistocene-Holocene age (Fadhilah & Prabowo, 2020; Zulkarnain, 2016). The climate of the area is humid and tropical and the mean annual precipitation in West Pasaman area is between 3500 and 4500 mm/year (Wilis, 2019). The Mw 6.1 earthquake hit West Sumatra on 25 February 2022. This event triggered several landslides in an area of 6 km$^2$, along the eastern and north-eastern flank of Talamau volcano.

### 2.9. Longchuan, China

The study area is located in the vicinity of Mibei village in Longchuan County, Guangdong Province, China with elevations between 180 and 600 m. The area has a subtropical monsoon climate, affected by frequent typhoons and rainstorms from May to October. The average annual precipitation ranges from 1300 to 2500 mm (Bai et al., 2021). The area is composed of Paleozoic completely weathered granite and Quaternary granite residual soil (Bai et al., 2021). Between 10 and 13 June 2019, an intense rainfall event, which was characterized by cumulative rainfall of 270 mm, triggered 327 shallow landslides between 300 and 400 m of altitudes and slopes ranging from 35 to 45° (Feng et al., 2022).

### 2.10. Hpa-An, Myanmar

The study area is located in Hpa-An district (central Kayin State, South Myanmar) in a tropical and monsoon area with a mean annual precipitation between 4500 and 5000 mm (Win Zin & Rutten, 2017) and elevations up to 1300 meters. The area is part of the Shan Plateau sequence, which includes low grade metamorphosed Precambian, Palezoic and Mesozoic sedimentary rocks (Jain & Banerjee, 2020). On 28–30 July 2018, Myanmar was hit by an extreme rainfall event which caused a flood along Bago river basin and triggered 992 landslides only in Kayin State (Amatya et al., 2022).

### 2.11. Kaikoura, New Zealand

The 2016 Kaikoura earthquake triggered more than 10,000 landslides in New Zealand, causing extensive damage and disrupting transportation routes. The landslides were complex and involved multiple failure mechanisms, including rockfalls, rock avalanches, and debris flows. The intense shaking and steep topography of the region contributed to the landslides. To reduce the potential impact of future earthquakes, it is crucial to improve understanding of

landslide mechanisms and develop effective early warning systems (Goda et al., 2020;
Massey et al., 2020; Wang et al., 2020).
2.12.        Uvira, Democratic Republic of Congo
The city of Uvira in the Democratic Republic of Congo experienced devastating landslides in
2020 due to heavy rainfall, poor land management practices, and the steep topography of the
region. These landslides caused significant damage to infrastructure and displaced thousands
of people. Landslides are a recurring hazard in the DRC, with an average of 100 occurring
annually, and climate change is expected to exacerbate the problem. Efforts to mitigate the
risk of landslides can include improved land use practices, early warning systems, and
infrastructure designed to withstand landslides. Taking a comprehensive approach is key to
minimizing the impact of landslides and protecting at-risk communities. (Mwene-Mbeja et al.,
2020; Kervyn et al., 2020; United Nations Office for Disaster Risk Reduction, 2020)
3.  High-Resolution Global landslide Detector Database (HR-GLDD)
3.1.  Data set description:
The dataset created in this study consists of images acquired from the PlanetScope satellites
(see table 1) and landslide inventories collected from the literature. For all the events,
landslides were manually delineated due to unavailability of existing inventories at high
reolution. PlanetScope is a constellation of approximately 130 satellites that acquire images
of the Earth daily with 3 meters of spatial resolution. The sensors acquire the images with 8
spectral bands: coastal blue (431 - 552 nm), blue (465 - 515 nm), green (547 - 583 nm), yellow
(600 - 620 nm), red (650 - 680 nm), red-edge (697 - 713 nm) and NIR (845 - 885 nm) (Planet
Team, 2019). PlanetScope imagery consists of surface reflectance values and 16 bits images.
The images from both sensors are orthorectified and radiometrically corrected by the providers
and we undertook the intrasensor harmonization process for the red, green, blue, and NIR
bands that is offered by PlanetScope.
The dataset was prepared using only the red, green, blue, and NIR bands. The pre-processing
phase was based on three steps: generation of binary masks, data sampling, and tiles
patching (Meena et al., 2022a).. We used manual image interpretation to manually delineate
landslide polygons.  First, the interpreted landslides polygons from each area were rasterized
using the Rasterio Python library into a binary mask, where "1" represents the landslides and
"0" the background. The satellite imagery, along with the mask was then sampled and patched
into a regular grid that yields patches of dimension 128 x 128 pixels, which correspond to 14.7
$km^2$ per patch. Since the imbalance between background area and landslides is strong, the
images that did not have any landslides pixel labelled were removed. The proportions for the
positive samples of landslides against the non-landslides are 9.96% and 90.04%, respectively.
Table 1 shows the number of tiles created for each area.
Table 1 - Number of tiles, satellite information and landslide statistics for each study area.

| Study Area | Satellite | Number of tiles | Study Area in $km^2$ | Number of landslides | Minimum Landslide area ($m^2$) | Maximum Landslide area ($m^2$) | Total Landslide area ($km^2$) |
|---|---|---|---|---|---|---|---|
| Kodagu India, 2018 | PlanetScope | 530 | 4033.62 | 343 | 276.23 | 581342.19 | 5.67 |

| | | | | | | | |
|---|---|---|---|---|---|---|---|
| Rolante Brazil, 2017 | PlanetScope | 33 | 24.62 | 113 | 381.76 | 81277.53 | 0.67 |
| Tiburon Peninsula, Haiti 2021 | PlanetScope | 461 | 130.85 | 1394 | 200.74 | 473696 | 8.24 |
| Rasuwa Nepal, 2017 | PlanetScope | 222 | 114.68 | 184 | 676.85 | 115567.96 | 2.45 |
| Hokkaido Japan, 2018 | PlanetScope | 159 | 50.17 | 715 | 237.76 | 48524.72 | 5.29 |
| Wenchuan China, 2017 | PlanetScope | 284 | 58.25 | 1415 | 23.78 | 98467.96 | 3.19 |
| Wenchuan China, 2018 | PlanetScope | 263 | 58.25 | 546 | 110.18 | 1289210.19 | 5.54 |
| Sumatra, Indonesia, 2022 | PlanetScope | 403 | 22.56 | 584 | 302.26 | 6206089.32 | 9.73 |
| Longchuan, China, 2019 | PlanetScope | 110 | 32.22 | 228 | 235.21 | 61163.17 | 0.73 |
| Hpa-An, Myanmar, 2018 | PlanetScope | 101 | 28.38 | 540 | 101.23 | 88044.20 | 0.97 |
| Porgera, Papua New Guinea, 2018 | PlanetScope | 725 | 304.94 | 491 | 262.65 | 259392.71 | 5.48 |
| Kaikoura, New Zealand, 2016 | PlanetScope | 287 | 150.75 | 246 | 676.67 | 165943.82 | 3.50 |
| Uvira, Democratic Republic of the Congo, 2020 | PlanetScope | 247 | 38.64 | 394 | 500.25 | 106094.52 | 1.61 |


3.2. Design of HR-GLDD
The performance evaluation of the study sites was carried out using metrics and trained using
five state-of-the-art U-Net like models, showcasing the capability and applicability of the High-

Resolution Global Landslide Detector Database (HR-GLDD). We used a total of ten geographically distinct study sites distributed globally, where landslide events were chosen including different triggering mechanisms such as five earthquake-induced and five rainfall-landslides-, we separately divide the patches into 60% for training, 20% for validation, and 20% for testing the model capabilities. All the sets are then mixed to create a unique dataset composed of equal percentages of patches.

We designed three scenarios to train, predict, and evaluate model performances in order to assess the robustness and applicability of the HR-GLDD. Primarily, we evaluate the model performances on the individual test sets (Meena et al., 2022a).. Secondly, we evaluate the performances of the models on the HR-GLDD test set. Moreover, finally, we test on two completely unseen recently occurred landslide events in Haiti 2021 and Indonesia 2022 (see figure 2).

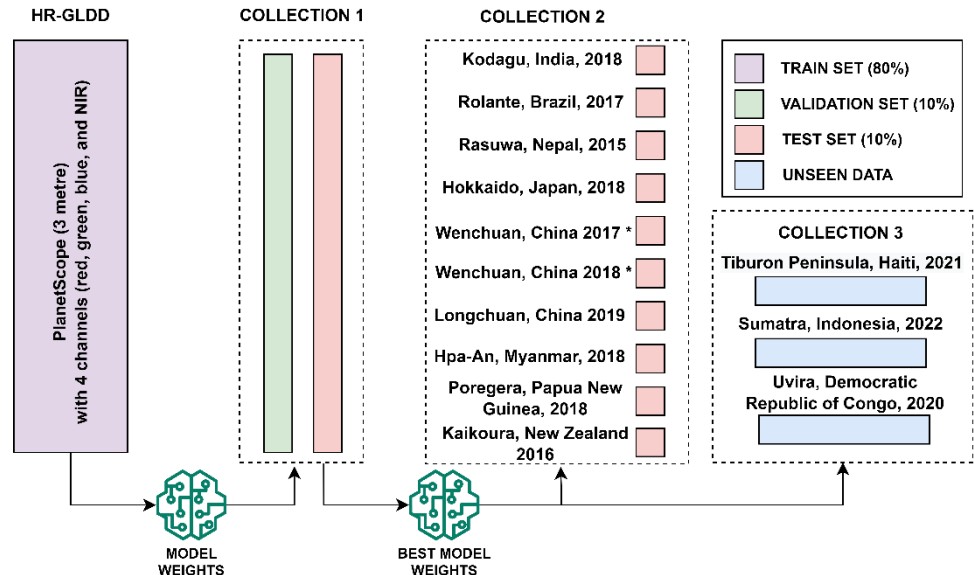

Figure 2: Schematic representation of the division of different components of HR-GLDD. Collection 1 refers to the test and validation data separated from the HR-GLDD. Collection 2 refers to the test dataset of individual sites. Collection 3 refers to the data from three recent events for testing purposes. Please note, while the Wenchuan event transpired in 2008, we've utilized images from a considerably later period, specifically those taken in 2017 and 2018. In an attempt to ensure the precision and accuracy of our analysis, we prioritized images with clearest, minimal cloud coverage.

## 4. Methodology

### 4.1. Model Architectures

The proposed dataset is evaluated through several state-of-the-art U-Net like Deep Learning segmentation models. In the past years, the U-Net (Abderrahim et al., 2020) has been used in several landslide detection applications which yield generally the most reliable results (Bhuyan et al., 2022; Meena et al., 2022c; Nava, Bhuyan, et al., 2022). Therefore, we decided to use it as a benchmark model when training on the proposed dataset. Moreover, several improved versions of the same are evaluated. We systematically trained the model using a variety of combinations of the hyper-parameters batch size (8, 16, 32, 64), learning rate (5e-4, 10e-4, 5e-5, 10e-5) and the number of filters of the first convolutional layer (8, 16, 32, 64). A description of the employed architectures is given in this section.

U-Net: This architecture has been utilized in various semantic segmentation applications, yielding generally outstanding results (Abderrahim et al., 2020). U-Net was employed initially in biomedical picture segmentation (Ronneberger et al., 2015). Low-level representations are captured by a contracting path (encoder), whereas a decoding path captures high-level representations. The encoding path consists of successive convolution blocks and is equivalent to a traditional CNN structure. Two convolutional layers with a 3 x 3 kernel size and a 2 x 2 max-pooling layer are present within every convolutional block. The rectified linear unit (ReLU) activation function is used to activate each convolutional layer (Agarap, 2018). A 2 x 2 max-pooling layer is added to the convolutional block's end in the encoder route to conduct non-linear downsampling, whereas, in the decoder path, a 2 x 2 upsampling layer takes its place. The upsampling layer is positioned right after a 3x3 convolutional layer (see figure S1). We refer to this combination as learnable upconvolution. We refer to this combination as learnable upconvolution.

Residual U-Net (Res U-Net): Res U-Net (Diakogiannis et al., 2020) follows the same U shape as U-Net, whereas here the above-explained convolutional blocks are replaced by residual blocks. This architecture's goal is to improve the learning capacities of the conventional U-Net as well as mitigate the gradient vanishing effect, especially when dealing with deep neural networks (such as U-Net) (see figure S2).

Attention U-Net and Attention Res U-Net: In the conventional U-Net as well as in the Res U-Net, cascading convolutions have been shown to provide false alerts for tiny objects with high form variability (Oktay et al., 2018). To select pertinent spatial information from low-level maps and therefore alleviate the problem, soft attention gates (AGs) are added (see figure S3, S4). The attention gates are built on skip connections, which actively inhibit activations in unnecessary areas, lowering the number of duplicated features (Abraham & Mefraz Khan, 2018).

Attention Deep Supervision Multi-Scale (ADSMS) U-Net: This architecture adopts the Attention U-Net structure, while, in addition, multi-scale image pyramid inputs are fed to the model, and a deep supervision strategy is applied (Abraham & Mefraz Khan, 2018). In practice, multi-scale inputs enable the model to gather that class data, which is more readily available at various sizes. This holds true for both background features and landslides. Lastly, where training data are few and networks are relatively shallow, deep supervision conducts a potent "regularization". More details about the deep supervision strategy used are available in the following section (see figure S5).

### 4.2. Model training

To train the DL models, we utilized Dice Loss ($\mathrm{DL}_c$) (Eq. 2) (Milletari et al., 2016) as the loss function:

$$\mathrm{DSC}_c = \frac{\sum_{i=1}^{N} p_{ic} g_{ic} + \epsilon}{\sum_{i=1}^{N} p_{ic} + g_{ic} + \epsilon} \tag{1}$$

Equation (1) illustrates a two-class Dice score coefficient (DSC) variation for the landslide class c, where $g_{ic} \in \{0,1\}$ and $p_{ic} \in [0,1]$ are the ground truth and predicted labels, respectively. Furthermore, the numerical stability is assured by avoiding division by zero, while N specifies the total number of picture pixels.

$$\mathrm{DL}_c = \sum 1 - \mathrm{DSC}_c \tag{2}$$

As an exception, in the ADSMS U-Net model, every high-dimensional feature representation is regulated by Focal Tversky Loss to avoid loss over-suppression, while the final output is

controlled by the conventional Tversky Loss (Eq. 4). The focal Tversky loss is a type of loss
function that focuses training on challenging cases, specifically those with a Tversky similarity
index ($TI_c$) (Eq. 3) of less than 0.5.
$$TI_c = \frac{\sum_{i=1}^{N} p_{ic} g_{ic} + \epsilon}{\sum_{i=1}^{N} p_{ic} g_{ic} + \alpha \sum_{i=1}^{N} p_{i\bar{c}} g_{ic} + \beta \sum_{i=1}^{N} p_{ic} g_{i\bar{c}} + \epsilon}$$    (3)

The Focal Tversky Loss ($FTL_c$) function incorporates the likelihoods of pixels belonging to the
landslide class ($p_{ic}$) and the background class ($p_{i\bar{c}}$) as well as the corresponding ground truth
labels ($g_{ic}$ and $g_{i\bar{c}}$). It is designed to handle significant class imbalances and can be adjusted
by modifying the α and β weights to prioritize recall.
The $FTL_c$ function is defined as follows:
$$FTL_c = \sum_c (1 - TI_c)^{1/\gamma}$$    (4)

where γ ranges between 1 and 3.
This deep supervision strategy, described in Lee et al., (2015), requires intermediate layers to
be semantically discriminative at all scales. Furthermore, it contributes to ensuring that the
attention unit has the power to change responses to a wide variety of visual foreground
material. This strategy is adopted from (Abraham & Mefraz Khan, 2018), who propose it along
with the ADSMS U-Net architecture. As the loss function optimizer, for all the models, we used
a stochastic gradient descent strategy based on an adaptive estimate of first- and second-
order moments (Adam), which is useful in problems with uncertain data and sparse gradients
(Kingma & Ba, 2015). The precision, recall, F1-score, and Intersection Over Union (IOU)
score, the most common accuracy evaluation measures for segmentation models, all of which
have been utilized in several landslide detection studies, were used to measure how well the
applied DL models performed in detecting landslides. The appropriate combinations of hyper-
parameters must be used while training such DL models in order to optimize the model and,
therefore, output the best results.
5.  Results
416        5.1. HR-GLDD evaluation results

The robustness and applicability of the HR-GLDD was tested using the best model weight.
We train and calibrate the models using the HR-GLDD. The best weighs for each model are
selected based on the model performances on the mixed test set of the HR-GLDD dataset.
After running the models on test dataset, batch size of 16 and Adam optimiser with learning
rate 5.00E-04 resulted in best model weight. To further evaluate the efficiency and
generalization capabilities of the models, we use the model on three unseen datasets to map
landslides in the two different geomorphological areas that were recently affected by multiple
landslide events. We chose the most recent events one occurred after Uvira, Democratic
Republic of Congo (DRC) heavy rainfall event of April 2020. Haiti earthquake in August 2021,
one in Sumatra, Indonesia after a heavy rainfall event of February 2022. A total of 247, 461
and 403 unseen image patches were evaluated for DRC, Haiti and Indonesia, respectively.
Experimental results for landslide detection by utilising the HR-GLDD are presented in Table
2. Overall, all the models performed consistently in collections 2 and 3. The F1-score
evaluation results for each test case of all the models demonstrate the applicability of the HR-
GLDD training dataset for landslide detection results. The average F1-score for HR-GLDD test
dataset (collection 1) across all the models was around 0.7045. Furthermore, the same was

observed in the individual test sites in collection 2. We also notice that the Precision and Recall are well balanced ranging between 0.6346-0.7661 and 0.6672-0.8121, respectively, indicating stable model predictions. In collection 3, the metrics reveal positive outcomes in terms of mapping the landslides following the respective events, with an average F1-score of 0.5562 for DRC, 0.7947 for Haiti and 0.8603 for Indonesia. The recall values are higher than precision values for all the models resulting in average F1-score of 0.7045 (see table 2). Higher values of recall in all models means that the models were able to identify landslide labelled pixels. However due to the use of only the optical bands, the spectral signatures of other similar features (such as riverbeds and flat barren areas) were labelled as landslides which result in false predictions, thereby, accounting for lower precision.

In figure 3 we chose a single image patch to showcase the predictions of the various models with respect to the referenced ground truth. Despite the differences in the spectral fingerprints of the satellite images for each study site and the events initiated by an earthquake or rainfall, the models were still capable of recognizing landslide features (see figure 4, 5 and 6). Particularly, we were able to map the recent events in DRC (2020), Haiti (2021) and Indonesia (2022).

Table 2: F1 scores of different DL models across sites and HR-GLDD test dataset along with three unseen test sites.

| | Study sites | U-NET | Res-U-NET | Attn-U-NET | Attn-res-Unet | ADSMS-U-NET |
|---|---|---|---|---|---|---|
| | Collection 1 (HR-GLDD Test) | 0.7904 | 0.6825 | 0.7446 | 0.6477 | 0.6576 |
| | | | | | | |
| Collection 2 | Kodagu, India, 2018 | 0.7674 | 0.6980 | 0.7628 | 0.6664 | 0.6796 |
| | Rolante, Brazil, 2017 | 0.7739 | 0.6913 | 0.6539 | 0.6830 | 0.6726 |
| | Rasuwa, Nepal, 2015 | 0.8972 | 0.8149 | 0.8419 | 0.7695 | 0.7976 |
| | Hokkaido, Japan, 2018 | 0.8159 | 0.7479 | 0.8124 | 0.7317 | 0.7552 |
| | Wenchuan, China, 2017 | 0.7781 | 0.6507 | 0.6981 | 0.6162 | 0.6739 |
| | Wenchuan, China, 2018 | 0.8077 | 0.6886 | 0.7295 | 0.6704 | 0.6557 |
| | Longchuan, China, 2019 | 0.6842 | 0.5076 | 0.5422 | 0.4829 | 0.4398 |
| | Hpa-An, Myanmar, 2018 | 0.8415 | 0.7861 | 0.7826 | 0.7405 | 0.7709 |
| | Porgera, Papua New Guinea, 2018 | 0.7515 | 0.6150 | 0.7568 | 0.6572 | 0.6261 |
| | Kaikoura, New Zealand, 2016 | 0.7496 | 0.5456 | 0.7335 | 0.4922 | 0.6494 |
| | Collection 3 | | | | | |

| | | | | | |
|---|---|---|---|---|---|
| *Sumatra, Indonesia, 2022* | 0.8832 | 0.8810 | 0.8232 | 0.8534 | 0.8608 |
| *Tiburon Peninsula, Haiti, 2021* | 0.8357 | 0.8055 | 0.7869 | 0.7648 | 0.7808 |
| *Uvira, Democratic Republic of the Congo, 2020* | 0.5937 | 0.5366 | 0.5682 | 0.5008 | 0.5819 |

454

455

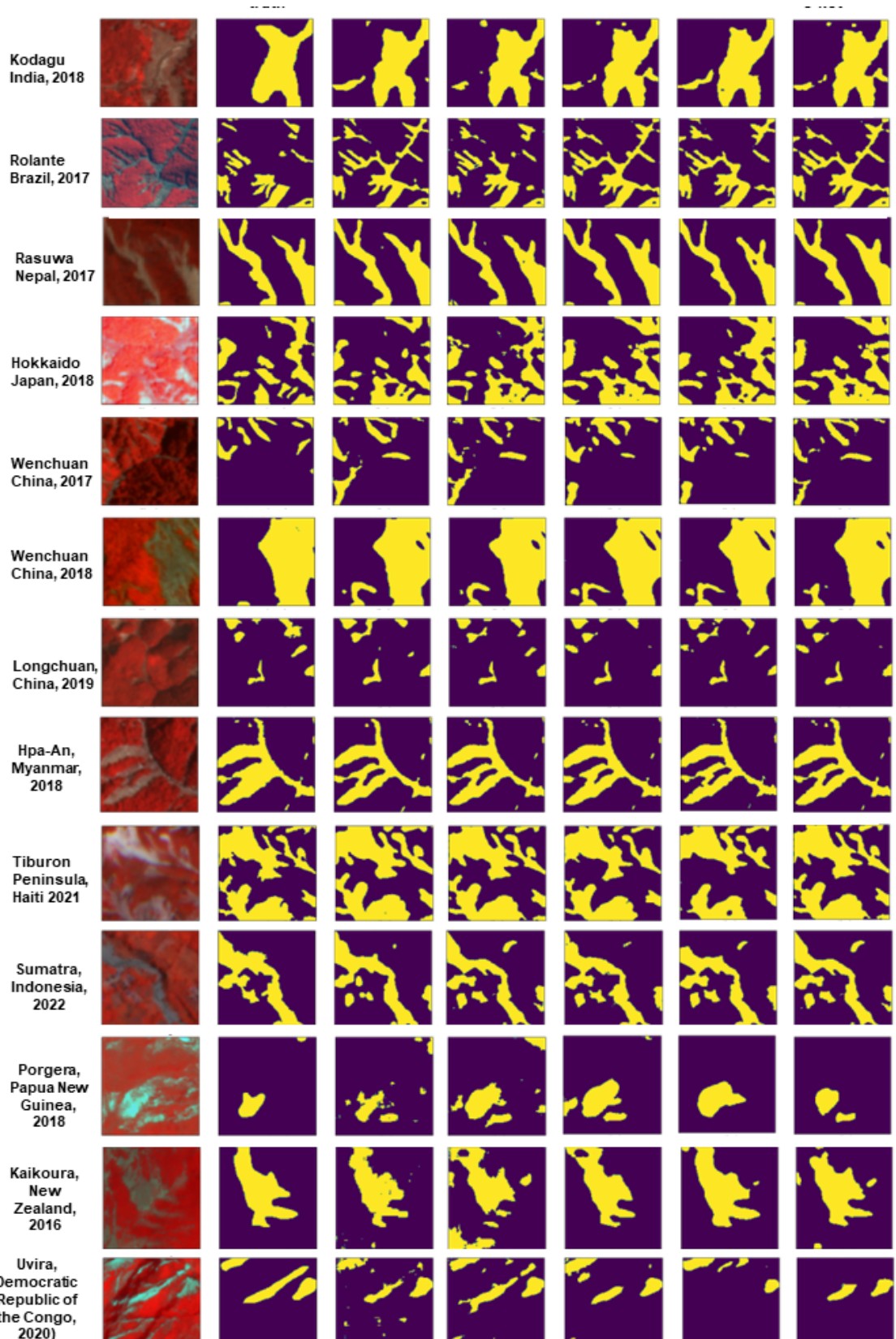

Figure 3: Landslide predictions made by the different DL models against the ground truth. The base image is shown as a false colour composite (FCC) to better visualize the scars of the landslides.

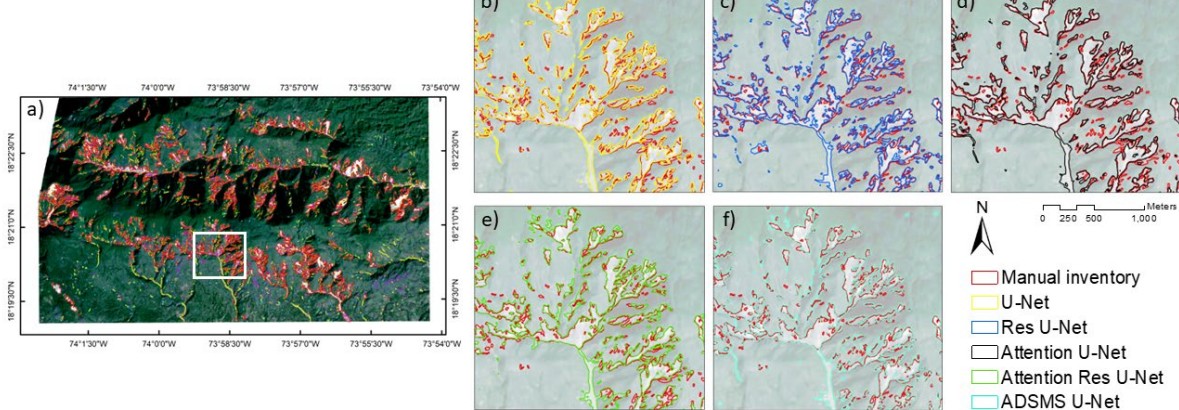

460

Figure 4: Comparison of ground truth landslides with predictions from the DL models for the unseen dataset of Haiti (We utilized various color coding schemes for the visualization of Deep Learning (DL)-based landslide detection results, allowing for a visual distinction between polygons generated from manual delineation)

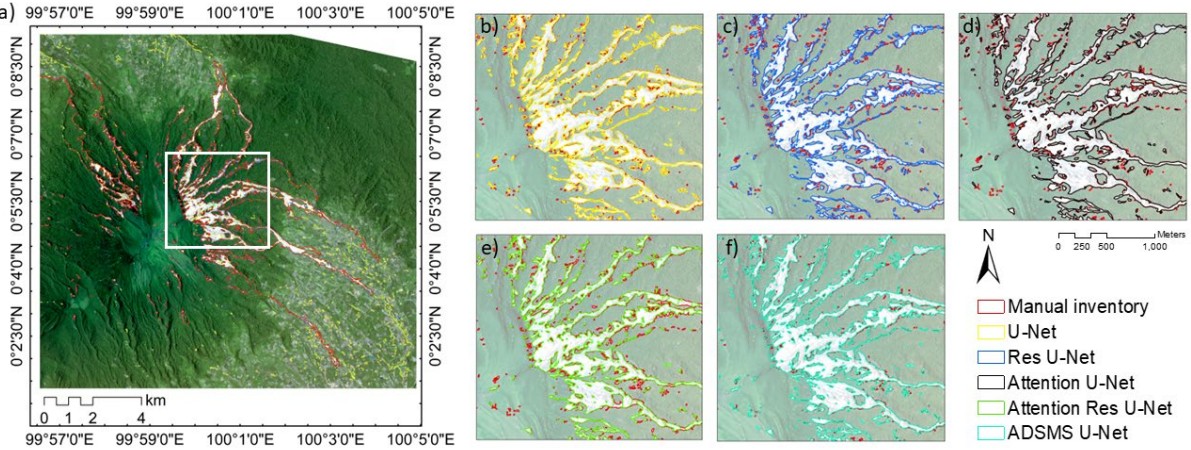

465

Figure 5: Comparison of ground truth landslides with predictions from the DL models for the unseen dataset of Indonesia Haiti (We utilized various color coding schemes for the visualization of Deep Learning (DL)-based landslide detection results, allowing for a visual distinction between polygons generated from manual delineation).

470

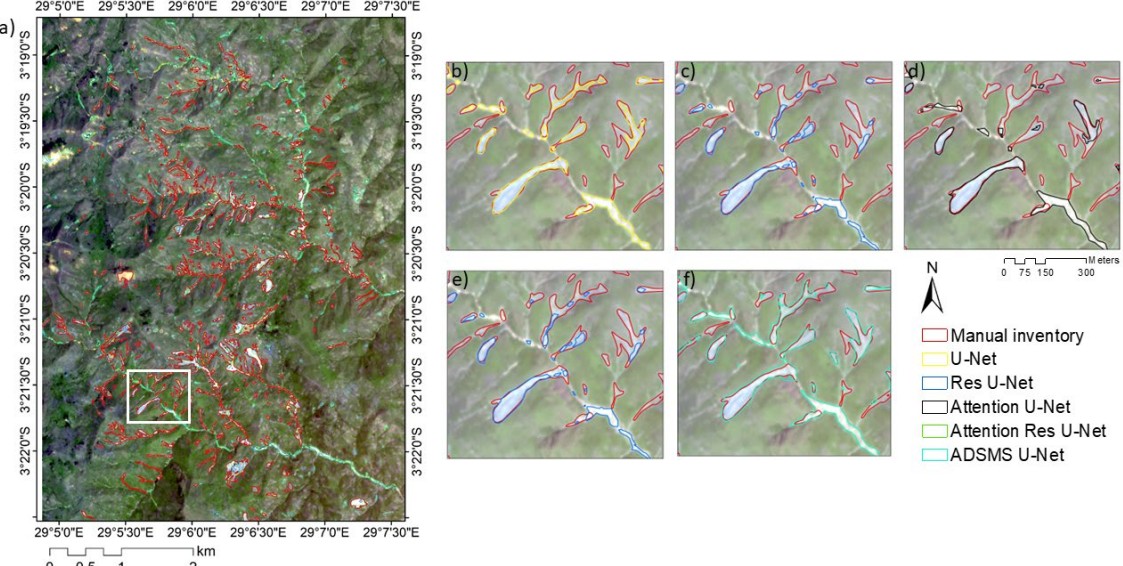

471

Figure 6: Comparison of ground truth landslides with predictions from the DL models for the unseen dataset of DRC (We utilized various color coding schemes for the visualization of Deep Learning (DL)-based landslide detection results, allowing for a visual distinction between polygons generated from manual delineation).

## 6. Discussions

### 6.1. Advantages of using HR images

The spatial resolution of Planet Scope imagery enables the detection of small size landslides that open access satellite missions like Sentinel and Landsat frequently miss due to their spatial and temporal resolution (Meena et al., 2021). Moreover, even though Sentinel-2 has additional spectral bands, the lack of improved spatial resolution inhibits precise boundary delineation and landslide localisation (Meena et al., 2022). The most prominent features of Planet Scope imagery, in addition to its competitive spatial resolution, are its daily temporal resolution and global coverage. Since the satellites have identical sensors, the imageries are orthorectified and image pre-processing are simplified and more accurate. Because Planet imagery provide global coverage, we may extend our study sites to new locations for generating more quality datasets that allow for a better model generalization.

### 6.2. Quality of HR-GLDD

The quality of any ML/DL model depends on the data that it is trained on, and the GLDD aims to meet this fundamental requirement. To our knowledge, no other quality data sets exist that can accommodate the wide range of landslide-triggering events and topographical diversity needed for efficient model training. As the GLDD is a strong collection of various landslide events caused both by rainfall and earthquakes. The GLDD is designed to calibrate models able to map new events that will occur in the future. The models investigated in our study gave promising and consistent results for two unseen datasets generated by completely different events, indicating a well-prepared, dependable, and resilient dataset. However, there are clear limitations with the GLDD that must be considered. These problems primarily stem from issues with manually delineated polygons and various uncertainties caused by satellite imagery. A number of different variables, including the mapping scale, the date, and the quality of the satellite imagery, affect how accurate an inventory is. The radiometric resolution and cloud coverage are additional variables that affect the generation of manual inventories. Additionally,

haze effect caused by instrument errors hinders model performances. Subjectivity in the
landslide polygon boundaries results from the amalgamation problem, which is caused by
elements like the investigators' level of experience and the goal of the study.
## 6.3. Significance of the HR-GLDD
A thorough hazard and risk framework is made possible by quality landslide inventories
however, the generating such inventories at large scales takes ample amount of time and
resources. This is where such automatic pipelines can truly shine at creating inventories which
can be used for the successive phases of a hazard and risk. Local, regional, and national
stakeholders may include such inventories into their risk reduction efforts thanks to the
availability of inventories produced automatically. Furthermore, this information may serve as
the foundation for a legal framework that implements landslide risk. A landslide risk reduction
plan is now more crucial than ever given the anticipated rise in worldwide landslide activity
brought on by climate change. Higher landslide activity is expected in the future due to a
number of factors, including an increase in the frequency and intensity of seismic events,
anthropogenic events, heavy precipitation events, rising ground water levels, storm surges,
and a general rise in relative sea level. Therefore, it is essential to comprehend the underlying
mechanisms of landslides better and create practical risk reduction techniques to save
people's lives and property.
## 6.4. Automated pipeline for HR-GLDD
At the moment, automated techniques are the only viable solution for mapping vast regions
with accuracy appropriate for operational objectives. Nonetheless, reliable, reproducible, and
accurate processes for automating landslide detection across huge data stacks are still
absent. As a result, many landslide-affected regions remain unmapped because 1) they are
challenging to map using standard methods, and 2) using high-resolution imagery is costly
and labour-intensive, with a substantial part of the mapping process dependent on human
judgment. By overcoming these challenges, automated pipelines that address these issues
can considerably reduce the requirement for human involvement and pave the way for the
development of reliable real-time mapping and monitoring of natural hazards at the continental
and global scales. Based on the quality of GLDD, reliability of automated pipelines and rapidly
growing availability of HR satellite imagery, we can realistically envision mapping of landslide
instances and contribute towards generating and updating landslide inventories at large-
scales, spatially and potentially, also temporally (Bhuyan et al., 2023).
Providing an expert-based, high-quality, and scientifically validated landslide inventory to
scientific communities is essential for frameworks of modelling, landslide prediction, machine
learning, and deep learning research. The GLDD dataset has been verified, which increases
the availability of much-needed training datasets for automated mapping algorithms. The
consistently long time taken to compile landslide inventories manually contrasts with the rise
in data accessible for landslide mapping. The development of technologies to successfully
automate the procedure is the future direction in landslide inventory mapping. The precedence
of quality dataset is noted in where they commented that the need for quality datasets will
provide a valuable resource for training and developing algorithms.
The current dataset is an excellent resource for training and developing future algorithms for
this purpose. Automated mapping methods, particularly when combined with publicly available
elevation models, can potentially improve our results in future investigations.
## 7. Conclusions

Mapping landslides through space is a challenging endeavour. Automated efforts for the same have been explored to some extent, but a transferrable method based on a robust dataset has not yet been investigated. In this paper, we propose a reliable dataset which can be employed by deep learning algorithms to detect new landslides accurately. The predictive capabilities demonstrate the usefulness and application of the dataset to map landslides at large scales. However, the model's predictability must be investigated further in order to identify particular problems to enhance the findings and predictive capabilities for more complicated landscapes. Overall, despite the limitations, the findings are promising, since it is the first time such a HR dataset has been created that caters to a transferable approach of mapping landslides at so many different geomorphological and geographical locations.

Data availability

The data, working codes and a document with metadata are freely available at https://doi.org/10.5281/zenodo.7189381 and https://github.com/kushanavbhuyan/HR-GLDD-A-Global-Landslide-Mapping-Data-Repository where data in the format of arrays and model configurations in the framework of TensorFlow as can be displayed and used for reproducibility of our results. We also submit the generated landslide inventories in the form of an Environmental Systems Research Institute (ESRI) shapefile. Modules for deep learning can be found at https://www.tensorflow.org/ and original satellite imageries can be found at https://www.planet.com/.

Code availability

Code used to produce data described in this manuscript, as well as to create figures and tables, can be accessed at https://github.com/kushanavbhuyan/HR-GLDD-A-Global-Landslide-Mapping-Data-Repository

Author contribution

All the authors contributed to equally to preparation of manuscript from data curation to review of final manuscript.

Competing interests

The authors declare that they have no conflict of interest.

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
