# Peer review of "HR-GLDD: A globally distributed dataset using generalized DL for rapid landslide mapping on HR satellite imagery"

_Earth System Science Data, 2022_

## Author Response (AR1)

**Reviewer #1**

General remark:

The paper tries to build a global high-resolution dataset for DL landslide detection. Though many tests have been conducted in the study, some major issues should be modified.

General Reply: We thank the reviewer for spending time in reading our manuscript and giving valuable insights, which we believe has improved the quality of our manuscript.

1.  The characteristics, triggers and conditions of landslides are often different from each other, even in a small area. In addition, the characteristics of landslides are often similar to other land covers. For instance, fresh landslides are like bare land; old landslides are often cover forests. For building a global dataset, 1st, more samples of different types, from hundreds of sites are needed; secondly, the differences between landslides and other land covers should be involved and studied. More importantly, the principle, and basic problems (e.g., types, influence factors, features) of landslide detection should be analyzed for collecting sufficient typical samples.

Reply: We thank the reviewer for their comment and respond to their query as follows.

The presence of landslides in remote sensing images is identified and mapped using segmentation models such as the ones we used in this work, which are commonly used for detecting and outlining the boundaries of landslides. Regarding the types of landslides, we agree that a crucial piece of information, in addition to their spatial location, is their failure mechanism. However, these models are not suitable for determining the specific failure mechanism of a landslide, such as whether it is a debris flow or a slide. This is because the specific physical processes that trigger a landslide, such as the type of soil or rock, the slope angle, and the presence of water, dictate the failure mechanisms. These factors cannot be inferred solely from the visual appearance of a landslide in a remote sensing image. Additional information, such as field observations, geotechnical data, and other remote sensing products, would be necessary to establish the failure mechanism of a landslide. Furthermore, even with the additional data, precisely identifying the failure mechanism of a landslide remains difficult due to the complex interplay of the different factors that contribute to landslides and the difficulty of effectively quantifying and measuring these components in the field. While segmentation models can be useful for identifying and mapping the existence of landslides, they cannot be used to determine the exact failure mechanism of a landslide. Furthermore, this is also beyond the scope of our manuscript. We are solely focusing on catering to the needs of the landslide detection community, where high-resolution data can be used by them for detecting new or recent landslides by means of quantity or quality data and/or means of transfer learning, as pointed out by Bhuyan et al. (2023) (https://www.nature.com/articles/s41598-022-27352-y).

We did put emphasis on the triggering factor and therefore, categorized them into earthquake and rainfall-induced landslides so that we had enough data from both triggering mechanisms.

Our study is also about gathering inventories that are distributed globally (as seen in Figure 1 of the manuscript) and they occur in varying geographical areas with varied land covers. Regarding the comment about introducing more samples of diverse types, we agree with this, but we also want to mention that this is an ongoing data documentation process. We spent months digitizing and gathering these data, and we will continue to do so moving forward by updating and adding more samples from more recent landslides. Like the Global Landslide Catalog, we will continuously update this database, but for now, we present the available data from our current repository in this manuscript.

2. In the paper, the work on building dataset is very little. The paper just produces image patches with landslides polygons from tens of sites, most of which are collected from other literatures. On the other hand, tests on different DL models occupied too much, which is not the focus of the study. It is not necessary to test so many UNet models, considering they display similar performance. For testing the performance of the dataset, more validation sites and different shallow or deep machine learning methods are suggested.

Reply: We agree with the reviewer that testing different DL models took up too much space. Therefore, we have moved the results of the improved U-Net architectures to the supplementary materials and included only the results of the conventional U-Net architecture in the main manuscript. This way, readers can use the results as a benchmark for comparing their own model performance. However, we disagree with the reviewer's comment that the work on building the dataset was minimal. In fact, we manually corrected and recreated the inventories of the selected areas using PlanetScope imagery, which ensured that the masks were reliable and precise, and that the landslide bodies in the masks matched those in the imagery. Furthermore, we believe that the dataset is reliably validated, as it was tested using two different test sets. The first test set was created by using an unseen percentage of each area used for training, and the second test set was created by predicting landslide locations in four completely new areas. Additionally, to further validate the dataset, we added two new cases in training the models and two for the unseen test dataset.

3. In introduction, previous works are not clearly and comprehensively stated. The significance of the study is unclear.

Reply: We disagree with this statement since we feel we have provided adequate information on the literature. We did mention the state of the art in terms of which/what models in the AI domain is used for landslide mapping. We have also mentioned about the current data sets provided for such training for example, Ghorbanzadeh et al. (2022) where they collected Sentinel-2 imageries for training purposes however, we also put forward the obvious limitations of such data, thereby highlighting the need of higher resolution data. We have also talked about the how different AI algorithms are used in a variety of geomorphological context, such as in Brazil, Nepal, Japan, and others to further emphasize that landslides are quite a ubiquitous issue, and many have tried different models to map them after they have occurred. We, therefore, believe that these are again, ample evidence of the state of the art. At the same time, we have included a few more examples from the recent literature to back up our claims/objectives. Please review the introductory section modifications.

4. Section 2 and section 3 should be merged.

Reply: Thank you for this suggestion. Yes, we have merged these two sections according to your suggestions. Please check the section.

5. Section 3.1 seems the frame work of the study, not a subsection of dataset.

Reply: We thank the reviewer for this comment. We agree in this regard and have moved the framework aspect to Section 4.1 and only left the description of the data in Section 3.

Reviewer #2

General remark:

In this study, a high resolution global distributed landslide dataset was generated to calibrate generalized DL models for event-based landslide segmentation. Promising results were reported in the delineation of landslide zones in two areas entirely not included in the 'calibration' set. Releasing the source code of the study on the GitHub platform is also valuable contribution of the study for the other researchers. However, there are some critical issues that must be mentioned and clarified for the publication of the manuscript.

Author Reply: We thank the reviewer for finding our paper valuable. We do understand the concerns of the reviewer and hence, we address them in the replies below. We hope that the reviewer finds the changes satisfactory and worthy for publication.

1. Although the different study areas used to sample the dataset are quite vary, the total number of training patches should be increased for better generalisation/model evaluation.

Author Reply: We thank the reviewer for this comment. We do agree with the argument and hence, we have added more study areas that increases the number of patches which caters to increasing the generalization capability of the models we tested. Please find them added under Section 2.1 for the regions of New Zealand, Papua New Guinea, and the Democratic Republic of Congo. You can also find the associated results of these regions in the Results section.

2. You use five U-Net like architectures. Also considering the literature, just the common one could be enough to evaluate the dataset and could be used as a benchmark for future research.

Author Reply: Yes, this is true however, just because the models are U-Net "like" does not mean that they behave all the same. Each model has their own strength and weakness. But the point was to see across a board of models, would the results still be similar? Because consistent results indicates that the data that we collected is robust and therefore, is model agnostic.

3. In the test you just use two unseen areas (Haiti and Indonesia). It would be better if you could include more unseen test site to better evaluate the generalization capabilities of such a dataset.

Author Reply: Thank you for this comment. Yes, we have added more study areas in the manuscript. Please find their descriptions in the study area section, and the results in the later results section.

4. What is the proportion between positive and negative samples? This should be inserted in the manuscript as it is an important information.

Author Reply: The proportions for the positive samples are 9.96% and 90.04% for non-landslides. We have added this information in Data Description section of the manuscript.

Minor concerns:

a. Why didn't you used further bands as NDVI and/or DEM derivates?

Author Reply: Thank you for this comment. Generally, yes, we would advise to use NDVI and/or DEM derivatives to improve results however, we would like to remind the reviewer that the point of the paper is to first generate a dataset that can be used for mapping landslides over different geographic settings. DEM and their derivatives can be used to identify but considering that the only open-source DEM available is the SRTM which is of 30 metres. Using this coarser resolution data with the high-resolution Planet Labs data would require matching the resolutions of the two data which would result in loss of information. Regarding the NDVI, we do use the NIR channel along with R, G, B channels and so, if required, a user can quite easily calculate the NDVI using the NIR and R channels. We do now realize that this could be a requested feature by end users so we will add a function in our GitHub repository that calculates the NDVI and concatenates this channel to the 4-band images. Thank you for the comment.

b. The U-Net architecture showed in Figure 3 is uncorrect. Please modify it.

Author Reply: Thank you for the comment. We have corrected Figure 3. Please find the corrections made in the respective Figure.

c. Lines 39-40: you could add more references about fatalities caused by landslides.

Author Reply: Thank you for the comment. We have added more references in accordance with your comment.

Reviewer #3

General remark:

In this study, high-resolution satellite images are used to generate landslide detection data set for landslide identification. And advanced deep learning models are assembled to test the portability and robustness of HR-GLDD. The author has conducted an interesting work to provide a toolbox to detect landslides based on the train on the data sets generated from 8 events landslides samples with different DL methods.

Author Reply: We thank the reviewer for taking their time in reading our manuscript and that they find our work interesting. We understand that there are certain concerns regarding the work and therefore, we addressed them all in our current version of the work. We hope you find the amended manuscript to your liking.

Specific Comments:

1.  Lines 16-18. In the abstract, the gaps in landslide mapping are not specific. Please describe briefly and specifically the contribution of this research.

Reply: Thank you for the comment. We have improved the gaps regarding landslide mapping in the abstract.

2.  HR-GLDD needs its full name for the first time in both abstract and in the main text.

Reply: We are sorry for this mistake. We have added the full name in the abstract and the main text.

3.  In the abstract, conclusions on the applicability of deep learning networks for landslide mapping are needed. Please describe which is the most suitable model for the landslides mapping among five most advanced deep learning models and the possible reasons and the applicable conditions.

Reply: We thank you for this comment. However, we prefer not to underline and give too much importance to the relative models' performance, since the focus of this research is on the potential of the proposed landslide segmentation dataset.

4. Landslide mapping, landslide detection and landslide inventory are mixed applied in this paper. Please unify the concepts as much as possible.

Reply: The reviewer is right. We should have mentioned what we mean by mapping, detection, and inventorying. So, we define them individually, but we unified them into one concept in the manuscript.

5. Lines 105-107, the introduction describes some landslide data sets used by experts, but it should be pointed out what are the shortcomings in the existing data sets and the advantages of the data set of this study, the differences between these 8 regions, and the characteristics of the data.

Reply: Thank you for this suggestion. At the moment, to our best knowledge, the only existing landslide segmentation dataset is the one created by Ghorbanzadeh et al., 2022. The dataset is created by sampling Sentinel-2 imagery (10 meters spatial resolution) from 4 different areas/events. The dataset we propose, instead, is sampled from 10 different areas/events and uses 3 meters spatial resolution imagery.

Sampling from more areas can provide a more diverse representation of both landslide and background classes, which can improve the robustness of the model when applied to different regions. Moreover, a dataset with more diversity is likely to generalize better to new unseen data than one with limited diversity, making it more suitable for real-world deployment. Sampling from 10 areas provide better coverage of the geographical region, reducing the risk of missing important features or patterns.

Higher spatial resolution imagery captures more detail, allowing for more accurate identification and segmentation of landslide features. It also allows to obtain a more detailed view, which can be useful to identify small landslides or details that may be difficult to see in lower resolution imagery. Moreover, it can provide more context for the location, helping to better understand the environment and the relationships between different objects and features. Therefore, the increased detail can result in improved accuracy when classifying features and objects, reducing the risk of misclassification.

6. In lines 242-243, the author removes the image without landslide, whether it will affect the prediction result of the real scene. Landslide occupies a small number of pixels, compared to the stable background, as shown in Table 1, "Study area in km2" and "Total Landslide area".

Reply: Thank you for the comment. In segmentation tasks the balance between two different classes is calculated upon the total number of pixels appertaining to each class. In this case (and often when we are dealing with landslide detection), in one image containing landslide instances, the pixels appertaining to the background are much more than the ones of landslides. Therefore, also by just keeping images containing at least one pixel labelled as landslides the overall dataset imbalance is still high. Therefore, if we would have kept all the images (also the ones without landslides) the models would not be able to learn properly, not even with the loss functions specific to deal with imbalanced sets. On the other hand, it is true that by omitting all the patches non containing at least a pixel of landslides, we feed to the model less background information. However, from experience, the best compromise is to adopt the solution we used in the paper, since when feeding also patches with 1 pixel labelled as landslides, we feed (for 128x128 patches) 16383 pixels of background and 1 of landslide. Therefore, we know that enough background information is fed to the models.

7. What does the yellow polygon in Figure 3 represent? Why is it represented as two green polygons in the last up-sampling, which is inconsistent with other layers? The upper subgraph in Figure 5 has the same problem, as Figure 6. In Figure 5, two subgraphs are not labelled.

Reply: Thank you for the comment. The yellow polygon in Fig. 3 is a mistake from our side. We have removed it. Same as the upper yellow polygon in Fig. 5 b. Figure 6 is correct, being the yellow polygon the output of the soft attention gates.

8. Line 347, the Dice loss equation is confusing.

Reply: Sorry, but we did not understand how the Dice loss equation is confusing? Could you please specify more explicitly? You can find more details about the Dice loss function in Lee et al., 2019.

9. Lines 352-353, Focal Tversky Loss was adopted in the ADSMS U-Net model. Focal Tversky Loss or Dice Loss was adopted for ADSMS U-Net model in this study. It's confusing me here.

Reply: Thank you for this comment. The ADSMS U-Net model because of deep supervision requires 4 loss functions (1 for each level). Abraham et al. 2019 proposes the model along with the Focal Tversky loss. As stated in the manuscript, "As an exception, in the ADSMS U-Net model, every high-dimensional feature representation is regulated by Focal Tversky Loss to avoid loss over-suppression, while the final output is controlled by the conventional Tversky Loss." meaning that the last three levels are regulated by the Focal Tversky, while the last (output level) is controlled by the Tversky loss. All the details of this model can be found in Abraham et al. 2019.

10. The author adopted the effect of high resolution to conduct experiments in 10 regions around the world. Here, I'm curious of the difference in the accuracy of deep learning landslide mapping among different types of landslides, or different image manifestations and different backgrounds, such as different vegetation cover, difference between foreground and background.

Reply: We thank the reviewer for this comment. Yes, this was something very important for us to understand if the data that we collected can be used to detect landslides with any type of segmentation models (in this case U-Net and their variant versions) and therefore, we chose these geographically diverse regions. However, we realised that perhaps we'd need to add more diverse regions and as such, we decided to add more complicated areas like New Zealand where not just the terrain and topographies, but also the spectral returns of the satellite imageries are very different. The landscape of New Zealand (specifically in the central eastern time) is characterized by mountainous terrain that bear very similar resemblance to landslide scars, and it is close to impossible to visually tell apart between the general slopes of the terrain and the landslides. We added this data to add more complexity to our dataset whereby, the vegetation cover and the rugged topography would help train the model more effectively in finding out landslides within such environments.

11. Lines 373-377, what's the best weights for each model? In this study, do you mix the data sets (i.e. landslides samples) or/and DL models?

Reply: Thank you for the questions. We did not insert in the manuscript the parameters associated with the best model weights. Now, these are added to the Supplementary materials.

The dataset is composed of a percentage of patches containing at least one landslide pixel samples from each study area. The same percentage and sampling strategy are used in all the sample sites, for all three sets (60% for the training set, 20% for the validation set, and 20% for the test set). Then these percentages from each area are concatenated together to compose one diverse training, one validation, and one test set.

Lastly, no, the DL models are not mixed, but trained and evaluated separately on the same training, validation, and test sets described above. Then a further validation step is performed on datasets sampled from completely new areas (and new events).

**REFERENCES**

Abraham, N., & Khan, N. M. (2019, April). A novel focal tversky loss function with improved attention u-net for lesion segmentation. In *2019 IEEE 16th international symposium on biomedical imaging (ISBI 2019)* (pp. 683-687). IEEE.

Ghorbanzadeh, O., Xu, Y., Ghamis, P., Kopp, M., & Kreil, D. (2022). Landslide4sense: Reference benchmark data and deep learning models for landslide detection. *arXiv preprint arXiv:2206.00515*.

Li, X., Sun, X., Meng, Y., Liang, J., Wu, F., & Li, J. (2019). Dice loss for data-imbalanced NLP tasks. *arXiv preprint arXiv:1911.02855*.

---

## Editor Decision (ED1)

*Dice loss application*

*It is about using the Dice loss application on binary coded image bands, the values of the 'ground truth' label of a pixel and the 'predicted label' of a pixel are either 0 or 1, representing whether the pixel refers to a landslide pixel or to the background. As a question, in the Dice coefficient application – would be the denominator the sum of total landslide pixels of both prediction and ground truth, and the numerator would be the sum of correctly predicted pixels? Here I am not sure, please go ahead in your manuscript with explaining the Dice loss application specifically linked to your application, (in very few sentences, for a good orientation of the reader related to your application).*

---

## Author Response (AR2)

**Public justification (visible to the public if the article is accepted and published)**:
Dear authors, and referees, many thanks for your contributions. Dear authors, thank you for your throughout revisions and adding additional data to your ESSD project.
There are minor editorial requirements for revision left before the manuscript can be published:

Dear Editor, Thank you for your thoughtful and constructive feedback on our manuscript. We appreciate your recognition of our efforts to improve the paper through comprehensive revisions and the addition of further data to HR-GLDD.

In general
i) an event year with a major landscape-wide landslide event and a region are needed for the identification of the specific data sets, accordingly, you can optimize the naming of the data sets in your manuscript text, tables, figures,
f.e., Figure 1 naming convention ((subregion) region and year) could be kept also for the naming in 'collections 2 and 3' in figure 2, also in the tables 1 (Papua New Guinea, New Zealand, Congo you could add the year of the event(s)), table 2 and figure 3.
f.e., manuscript text, chapter 2 L137 '…we selected 8 regions…' and number of events-
Accordingly,

Thank you for the comment; we have made the suggested changes in the abstract and also throughout the text. See lines 23- 34 in the abstract and figures 2 and 3, and also tables 1 and 2.

ii) you are introducing in chapter 2 two times Papua New Guinea in 2,1 and 2.11 - 2.1 Papua New Guinea, describes a 2018 landslide event and 2.11. Porgera, Papua New Guinea, describes a 2018 event –please combine Papua New Guinea 2.1 and 2.11

Please see updated section 2.1.

iii) f.e., in abstract L30 and chapter 3.2, L309 You name two recent landslide events, or two regions of unseen data – it should be three that you put in for validation? Please carefully check your manuscript

Please see the change in line# 34 in abstract.

iv) Github and Dataset publication are well set up,
abstract text and figures would need to be updated for the final manuscript publication, e.g. also putting in the technical information on PlanetScope, used bands and binary mask in the Zenodo abstract

Please see line #23-34 in abstract.

v) in ESSD, data sets need to be cited in the reference list:
Sansar Raj Meena, Lorenzo Nava, Kushanav Bhuyan, Silvia Puliero, Lucas Pedrosa Soares, Helen Cristina Dias, Mario Floris, & Filippo Catani. (2022). HR-GLDD: A globally distributed high resolution landslide dataset [Data set]. Zenodo. https://doi.org/10.5281/zenodo.7189381

we have cited the dataset in the reference list.

vi) ESSD requires a detailed description of the data sets, please add some more technical information on source and characteristics already in the abstract

HR-GLDD), a high resolution (HR) satellite dataset (PlanetScope, 3 m pixel resolution) for landslide mapping landslide instances from ten different physiographical regions globally: South and South-East Asia, East Asia, South America, and Central America. The dataset contains areas of five rainfall triggered and five earthquake triggered multiple landslide events that occurred in varying geomorphological and topographical regions in the form of standardized image patches containing four PlanetScope Image bands (red, green, blue, and NIR) and a binary mask for landslide detection. Figure 2: you could put this data collection information in the empty box 'HR-GLDD'

Please check abstract and figure 2.

Specific

i) chapter 3 L282 The images from both sensors are orthorectified and radiometrically corrected by the providers.
PlanetScope Image data can come from 3 different sensor types: PS2, PS2.SD, PSB.SD
Please mention if, did you undertake the intrasensor harmonization process for the specific bands that is offered by PlanetScope?

Please see lines# 286-287, we have added "we undertook the intrasensor harmonization process for the red, green, blue, and NIR bands that is offered by PlanetScope".

ii) chapter 3, delineation of the landslide areas polygons, how did you manually delineate the landslides to create the binary masks, e.g. did you use quasi true RGB or False Colour Image? Please add a sentence

Please see line #290.

iii) Figure 4, please add an explanation of the color coding of your visualization of the binary masks, e.g., yellow = Landslide, black = background.

We have added the explanation in figure 4, 5 and 6.

iv) We agree with Reviewer 3 concerns that the presentation of the Dice Loss equation in its current form is confusing, e.g. you are using the Calculus mathemical language, i for imaginary number, and N indexed by I, however, the equation in the manuscript goes on in the form of computer scripts, please bring the Dice coefficient equation, or extended as Dice loss equation into the form of an integral equation with denominators and numerators, (Lee et al. 2019, Milletari et al., 2016), see also further questions in attachment.

We are sorry for the wrong dice loss equation, it was result of formatting issue in Microsoft word, we have updated the section 4.2 model training.